# Identification and Validation of a Novel Ferroptotic Prognostic Genes-Based Signature of Clear Cell Renal Cell Carcinoma

**DOI:** 10.3390/cancers14194690

**Published:** 2022-09-27

**Authors:** Zhiyuan Shi, Jianzhong Zheng, Qing Liang, Yankuo Liu, Yi Yang, Rui Wang, Mingshan Wang, Qian Zhang, Zuodong Xuan, Huimin Sun, Kejia Wang, Chen Shao

**Affiliations:** 1Department of Urology, Xiang’an Hospital of Xiamen University, School of Medicine, Xiamen University, Xiamen 361101, China; 2Fujian Provincial Key Laboratory of Organ and Tissue Regeneration, Organ Transplantation Institute of Xiamen University, School of Medicine, Xiamen University, Xiamen 361101, China; 3Xiamen Key Laboratory of Regeneration Medicine, Organ Transplantation Institute of Xiamen University, School of Medicine, Xiamen University, Xiamen 361101, China; 4Central Laboratory, Xiang’an Hospital of Xiamen University, School of Medicine, Xiamen University, Xiamen 361101, China

**Keywords:** clear cell renal cell carcinoma, ferroptosis, prognostic model, TCGA, E-MTAB-1980, GLS2

## Abstract

**Simple Summary:**

Clear cell renal cell carcinoma (ccRCC) is one of the leading types of kidney malignancy and is closely related to ferroptosis that is an iron-dependent regulated cell death with lipid peroxide accumulation. A signature of nine ferroptotic genes was identified as an independent prognostic factor via construction in The Cancer Genome Atlas (TCGA) database and validation in the ArrayExpress database. This signature could successfully divide patients into low- and high-risk groups to predict survival rate. Compared with the other eight genes, glutaminase 2 (GLS2) played a crucial role during erastin-induced ferroptosis in ACHN and Caki-1 cells. It was discovered for the first time that GLS2 might be a ferroptotic suppressor in ccRCC.

**Abstract:**

Renal cell carcinoma (RCC), as one of the primary urological malignant neoplasms, shows poor survival, and the leading pathological type of RCC is clear cell RCC (ccRCC). Differing from other cell deaths (such as apoptosis, necroptosis, pyroptosis, and autophagy), ferroptosis is characterized by iron-dependence, polyunsaturated fatty acid oxidization, and lipid peroxide accumulation. We analyzed the ferroptosis database (FerrDb V2), Gene Expression Omnibus database, The Cancer Genome Atlas database, and the ArrayExpress database. Nine genes that were differentially expressed and related to prognosis were involved in the ferroptotic prognostic model via the least absolute shrinkage and selection operator Cox regression analysis, which was established in ccRCC patients from the kidney renal clear cell carcinoma (KIRC) cohort in TCGA database, and validated in ccRCC patients from the E-MTAB-1980 cohort in the ArrayExpress database. The signature could be an independent prognostic factor for ccRCC, and high-risk patients showed worse overall survival. The Gene Ontology and Kyoto Encyclopedia of Genes and Genomes were utilized to investigate the potential mechanisms. The nine genes in ccRCC cells with erastin or RSL3 treatment were validated to find the crucial gene. The glutaminase 2 (GLS2) gene was upregulated during ferroptosis in ccRCC cells, and cells with GLS2 shRNA displayed lower survival, a lower glutathione level, and a high lipid peroxide level, which illustrated that GLS2 might be a ferroptotic suppressor in ccRCC.

## 1. Introduction

Renal cell carcinoma (RCC) is one of the main urological malignant tumors, and there were more than 430 thousand new cases and nearly 180 thousand deaths worldwide in 2020 [1]. The ratio of men to women diagnosed with RCC was nearly 1.7:1 and 90% of cases of RCC were in patients over 50 years of age [2]. The risk factors include hypertension, tobacco smoking, and obesity [3]. The leading pathological type of RCC is clear cell RCC (ccRCC) that accounts for approximately 75% [3], and the percentage of metastatic ccRCC is 83–88% [4]. Patients diagnosed at an early stage possess a better survival rate of 80–90% [5], however, only 30% of patients could be diagnosed according to symptoms and 40% of patients whose 5-year survival rate is under 11.2% develop distant metastases [6,7]. In summary, early diagnosis and precise management of ccRCC is significant because of their correlation with prognosis.

Ferroptosis, as one of the regulated cell deaths, was a term coined by Scott J. Dixon and Brent R. Stockwell in 2012 [8]. Differing from other cell deaths (such as apoptosis, necroptosis, pyroptosis, and autophagy), ferroptosis is dependent on the oxidization of polyunsaturated fatty acid (PUFA) and subsequent lipid peroxide accumulation [9]. When ferroptosis occurs in cells, there are several distinguishing features including the rupture of cellular membranes and a series of mitochondrial changes (mitochondria shrink, higher mitochondrial membrane density, reduced mitochondria cristae, and outer mitochondrial membrane rupture), however, there is no obvious change in the nucleus [10]. The molecular compound erastin and Ras selective lethal 3 (RSL3) could induce ferroptosis through different mechanisms. The erastin could inhibit solute carrier family 7 member A11 (SLC7A11) to prevent the import of cystine, which limits glutathione (GSH) synthesis [8]. Erastin could also bind with voltage-dependent anion channel 2 (VDAC2) to produce reactive oxygen species (ROS) in a NADH-dependent manner, which induces mitochondrial damage [11]. Different from erastin, RSL3 could directly repress the enzymatic activity of GSH peroxidase 4 (GPX4) that reduces lipid peroxides to corresponding alcohols via the inactivation of its enzymatic active site, and the reduction of GPX4 needs GSH to recover this site [12,13,14]. On the other hand, the intracellular level of iron is also significant because it plays a crucial role in participating in lipid peroxidation. Fe^2+^ could generate the hydroxyl radical (HO•) via the Fenton reaction [15]. The HO•, as the most active ROS, could initiate nonenzymatic lipid peroxidation including PUFA oxidation [10]. Therefore, the metabolism progress associated with GSH synthesis, iron metabolism, and lipid peroxidation might affect ferroptosis, especially in protein-coding genes, and the expression of some ferroptotic genes might change to reduce ferroptosis [16].

In previous research, several studies have explored prognostic markers in ccRCC, and a few ferroptosis-related prognostic models have been constructed [17,18,19,20,21,22]. However, previous studies only took into consideration 60 genes from references [23,24]. In this study, we made full use of the ferroptosis database (FerrDb V2), Gene Expression Omnibus (GEO) database, The Cancer Genome Atlas (TCGA) database, and the ArrayExpress database to explore the ferroptotic prognostic gene-based model of ccRCC. From the twenty genes that were differentially expressed and associated with prognosis, nine genes were involved in the ferroptotic prognostic model (FPM) via the least absolute shrinkage and selection operator (LASSO) Cox regression analysis, which was established in ccRCC patients from the kidney renal clear cell carcinoma (KIRC) cohort in TCGA database, used to establish a nomogram for conducting individualized prognosis assessment, and validated in ccRCC patients from the E-MTAB-1980 cohort in the ArrayExpress database. The signature could be an independent prognostic factor for ccRCC, and patients in high-risk groups showed worse prognosis. Then the Gene Ontology (GO) and Kyoto Encyclopedia of Genes and Genomes (KEGG) were employed to explore the potential mechanisms, and we tried to validate the nine genes in ACHN and Caki-1 cells with erastin or RSL3 treatment. The glutaminase 2 (GLS2) gene was upregulated during ferroptosis in ccRCC cells, and the cells treated with GLS2 shRNA displayed lower survival, a lower glutathione (GSH) level, and a high lipid peroxide level. Here, we discovered for the first time that GLS2 might be a ferroptotic suppressor in ccRCC.

## 2. Materials and Methods

### 2.1. Collection and Processing of Data

The GSE53757 (72 normal kidney tissues and 72 ccRCC tissues), GSE66272 (27 normal kidney tissues and 27 ccRCC tissues), and GSE71963 (16 normal kidney tissues and 32 ccRCC tissues) databases were downloaded from the GEO database (https://www.ncbi.nlm.nih.gov/geo/, accessed on 31 May 2022). The ferroptosis-related genes (including drivers, suppressors, and markers) were obtained from the FerrDb V2 database (http://www.zhounan.org/ferrdb, accessed on 31 May 2022). The datasets of ccRCC were obtained from the KIRC cohort (72 normal kidney samples and 539 ccRCC samples) in TCGA database (https://portal.gdc.cancer.gov, accessed on 31 May 2022) and the E-MTAB-1980 cohort (101 ccRCC samples) in the ArrayExpress database (https://www.ebi.ac.uk/arrayexpress, accessed on 31 May 2022). The proteomic dataset of ccRCC was downloaded from the Clinical Proteomic Tumor Analysis Consortium (CPTAC, https://proteomic.datacommons.cancer.gov/pdc, accessed on 31 May 2022). 

The following data processing was handled by R language software. The gene expression of the above databases was normalized with the “edgeR” package [25]. The differentially expressed genes (DEGs) between normal kidney tissue and ccRCC tissue in the GSE53757, GSE66272, GSE71963, and KIRC cohort were obtained using the “limma” package [26], and DEGs were selected with the |log_2_ (FC)| > 1 and the false discovery rate (FDR) < 0.05. The tumor prognostic genes (TPGs) were obtained using the “survival” package and identified by univariate Cox analysis (*p* < 0.05) of overall survival (OS) [27].

### 2.2. Establishment and Validation of the FPM

The LASSO Cox regression analysis was used to select ferroptotic prognostic DEGs (FPEDGs) using the “glmnet” package, and the optimal lambda (λ) was identified as the optimal value through a tenfold cross-validation process [28,29]. The risk score was computed by summarizing the product of each normalized FPEDG expression and its corresponding multivariate Cox regression coefficient (β). In the KIRC cohort, the risk score of every patient was computed using the above formula, and ccRCC patients were divided into two groups (low-risk group and high-risk group) by the media risk score. In the E-MTAB-1980 cohort, ccRCC patients were divided into two groups (low-risk group and high-risk group) according to the media risk score of the KIRC cohort. The distribution patterns of patients were performed by uniform manifold approximation (UMAP) using the “umap” package and t-distributed stochastic neighbor embedding (t-SNE) with “Rtsne” package. The “survminer” package was employed to conduct Kaplan–Meier (K–M) survival analysis. The “time” ROC package was employed to conduct time-dependent receiver operating characteristic (ROC) analysis.

### 2.3. Independent Prognostic Value of FPM

The risk score and clinical factors including age, gender, grade, and stage were analyzed via Pearson’s chi-square test and displayed using a heatmap. The univariate/multivariate Cox regression analysis was employed to estimate the independent prognostic value of FPM and traditional clinical characteristics, and the results were summarized using hazard ratios (HRs) and 95% confidence intervals (CIs).

### 2.4. Establishment and Validation of the Nomogram

According to the results of the multivariate Cox regression analysis, the factors were used to establish a nomogram for predicting survival rates using the “rms” package and the “survival” package. Time-dependent ROC analysis was utilized to evaluate the predictive performance of FPM using the “time” ROC package. Calibration curves were utilized to estimate the consistency between actual survival rates and predicted survival rates.

### 2.5. Molecular Functional Analysis

The GO and KEGG of the nine genes were displayed using the “clusterProfiler” package to screen the potential biological processes (BPs), cellular components (CCs), molecular functions (MFs), and pathways. These results were presented using the “ggplot2” package and the “G0plot” package [30].

### 2.6. Regents and Assay Kits

Fetal bovine serum (10100147), DMEM high glucose media (11965092), RPMI 1640 media (11875119), and penicillin-streptomycin (15140163) were purchased from Gibco (Foster, CA, USA). Erastin (S7242), ferrostatin-1 (Fer-1, S7243), liproxstatin-1 (Lip-1, S7699), hydroxychloroquine sulfate (HCQ, S4430), necrosulfonamide (NSA, S8251), Z-VAD-FMK (Z-VAD, S7023), and RSL3 (S8155) were purchased from Selleck (Houston, TX, USA). GLS2 rabbit polyclonal antibody (ab113509) was purchased from Abcam (Cambridge, UK). Actin mouse monoclonal antibody (A5441) and monobromobimane (MBB, B4380) were purchased from Sigma-Aldrich (Saint Louis, MO, USA). MTT (S19063) was purchased from Shanghai yuanye Bio-Technology Co., Ltd. (Shanghai, China). Lipofectamine 2000 (11668019), Hoechst 33,342 (H1398) and BODIPY 665/676 (B3932) were purchased from Thermo (Waltham, MA, USA). The SPARKeasy cellular RNA extraction kit (AC0205) was bought from Shandong Sparkjade Biotechnology Co., Ltd. (Jinan, China). The Evo M-MLV RT Kit (AG11711) was purchased from Accurate Biotechnology (Hunan) Co., Ltd. (Changsha, China). The puromycin (60210ES25) and Hieff^®^qPCR SYBR Green Master Mix (11201ES08) were purchased from Yeasen Biotechnology (Shanghai) Co., Ltd. (Shanghai, China). RIPA buffer (R0010), a GSH assay kit (BC1175) and a malondialdehyde (MDA) assay kit (BC0025) were bought from Beijing Solarbio Science & Technology Co., Ltd. (Beijing, China).

### 2.7. Cell lines and Cell Culture

ACHN and Caik-1 cells were provided by Cell Bank/Stem Cell Bank, Chinese Academy of Sciences. ACHN was cultivated in DMEM high glucose media with 1% penicillin-streptomycin and 10% fetal bovine serum, whereas Caki-1 cells were cultivated in RPMI 1640 media with 1% penicillin-streptomycin and 10% fetal bovine serum. All cells were cultured in a humidified incubator at 37 °C with 5% CO_2_.

### 2.8. Cell Viability Assay

A total of 5 × 10^3^ ACHN or Caik-1 cells were seeded into each well of a 96-well plate and were incubated overnight, then the cells were respectively treated with various concentrations of erastin (12 h) or RSL3 (6 h) with or without Fer-1 [1 μmol/L (μM)], Lip-1 (1 μM), HCQ (20 μM), NSA (1 μM), and Z-VAD (20 μM) [31]. Then, 5 mg/mL MTT was added to each well and incubated for 4 h at 37 °C. Then, the medium was carefully removed from the wells and 150 μL DMSO was added into each well. After shaking for 10 min, the 96-well plate was put into a multiskan sky high microplate reader (Thermo, Waltham, MA, USA) and the absorbance was detected at 570 nm wavelength.

### 2.9. Real Time-PCR Assay

Total RNA from cells were extracted by using a cellular RNA extraction kit, and reverse transcribed into cDNA by using the cDNA synthesis kit [32]. The real time PCR was performed using Bio-Rad CFX96 Real-time PCR systems (Bio-Rad, Hercules, CA, USA). The result was calculated by the comparative Ct method. All primers were designed by Primer Premier 6 and synthesized by Sangon Biotech (Shanghai, China). The primer sequences are shown in Appendix A.

### 2.10. Lentiviral Infection

The GLS2 shRNA sequence was taken from references and synthesized by Shanghai Genechem. Viral vector and packaging vectors were transfected into HEK293T cells using Lipofectamine 2000. The medium was replaced after 6 h, and viral particles were harvested after 24 h. After ACHN and Caki-1 cells had been infected for 24 h, the medium was replaced, and cells were cultured for another 48 h. Then puromycin (2.0 μg/mL) was employed to select cell lines.

### 2.11. Western Blot

When cells were cultured to 90% confluence, they were harvested and lysed with RIPA buffer containing protease inhibitor cocktail (HY-K0010, MedChemExpress, Monmouth Junction, NJ, USA). The protein concentration was measured by using a BCA protein assay kit (23227, Thermo). Then, samples were separated by 10% SDS polyacrylamide gel, transferred onto polyvinylidene difluoride (PVDF) membrane (WBKLS0500, Millipore, Billerica, MA, USA), blocked with 5% skimmed milk for 1 h at room temperature, and blotted with the corresponding antibodies (actin 1:10,000; GLS2, 1:1000) in 5% skimmed milk overnight at 4 °C. The PVDF membranes were washed with TBST three times and incubated with HRP-conjugated secondary antibody for 1 h at room temperature. After being washed with TBST three times, the PVDF membranes were detected by enhanced chemiluminescence reagents (WBKLS0050, Millipore, Billerica, MA, USA) with C300 (Azure Biosystems, Dublin, CA, USA).

### 2.12. Detection of MDA and GSH Level

The levels of MDA and GSH in cells were measured using MDA and GSH assay kits and detected using a multiskan sky high microplate reader (Thermo, Waltham, MA, USA).

### 2.13. Lipid Peroxidation Assay

The cells were seeded on glass-bottomed culture dishes and incubated for 24 h. The cells were incubated with DMSO, erastin or erastin+Lip-1 solutions for 12 h and washed with PBS. Next, the cells were treated with 10 μM BODIPY 665/676 dissolved in PBS for 30 min at 37 °C and washed with PBS. Then the cells were treated with 2 μg/mL Hoechst 33,342 for 15 min at room temperature, washed with PBS, and examined using a Zeiss LSM 880+ Airyscan confocal microscope (Zeiss, Oberkochen, Germany).

### 2.14. MBB Staining

MBB dissolved in PBS was used to treat cells for 15 min at 37 °C after the removal of culture medium [33]. They were photographed using an Olympus IX51 inverted fluorescence microscope (Olympus, Tokyo, Japan).

### 2.15. Statistical Analysis

All data were displayed by mean plus or minus standard deviation. Statistical analysis was managed using Prism 9 and SPSS 13. The value of *p* < 0.05 was considered significant (* *p* < 0.05, ** *p* < 0.01, *** *p* < 0.001).

## 3. Results

### 3.1. Identification of the FPDEGs

In this study (Figure 1), we identified 1519 DEGs (Appendix A) from three GEO databases (GSE53757, GSE66272 and GSE71963), 449 genes (Appendix A) from the FerrDb V2 database, and 5865 DEGs (Appendix A) from TCGA database, and then 41 significant DEGs (Appendix A) were obtained. Next, we identified 207 tumor prognostic genes (TPGs) (Appendix A) from the FerrDb V2 database and TCGA database. Finally, we harvested 20 FPEDGs that contained 11 upregulated genes and nine downregulated genes between normal tissues and ccRCC tissues in Figure 2A. The expression and HR (95% CI) of 20 FPEDGs in TCGA are displayed in Figure 2B,C.

### 3.2. Construction of the FPM in TCGA

The 20 FPEDGs were subjected to LASSO Cox regression analysis based on OS to screen key genes among FPEDGs (Figure 3A,B). A nine-gene signature with DPEP1, NOX4, MT1G, GLS2, GLRX5, TIMP1, CA9, CDCA3, and CYBB was identified in the KIRC cohort based on the optimal λ, and their respective relative coefficients were calculated to establish the FPM. The risk score of each patient was computed using the following formula: Risk score = (−0.12661 × expression of DPEP1) + (−0.03457 × expression of NOX4) + (0.06429 × expression of MT1G) + (−0.06666 × expression of GLS2) + (−0.31540 × expression of GLRX5) + (0.21071 × expression of TIMP1) + (−0.17067 × expression of CA9) + (0.56213 × expression of CDCA3) + (−0.04120 × expression of CYBB).

After checking the clinical data and gene expression of the patients in TCGA, we deleted 13 cases without gene expression profiles, complete clinical data or patients with 0 days. Based on the media risk score, patients (*n* = 526) were divided into a low-risk group (*n* = 263) and a high-risk group (*n* = 263), and patients in the high-risk group possessed high mortality (Figure 3C). The two groups of ccRCC patients could be well distributed into two sets by using UMAP and t-SNE analysis (Figure 3D,E). The K–M survival curves of the two groups showed that patients in the high-risk group had a worse survival rate when compared with their counterparts (Figure 3F). Furthermore, the time-dependent ROC analysis was utilized to show the prognostic value and predictive performance of the FPM, and the area under curve (AUC) reached 0.751 at 1 year, 0.732 at 3 years, and 0.748 at 5 years (Figure 3G), which illustrated that the FPM could be a suitable prognostic predictor.

### 3.3. FPM Could Be a Well Independent Prognostic Factor of ccRCC

The heatmap based on increasing risk score showed the relationship between risk group, basic clinical information, pathological feature, and nine-gene expression (Figure 4A). Based on Pearson’s chi square test, the risk group had noticeable correlativity with the tumor grade (*p* < 0.001), tumor stage (*p* < 0.001), and patient status (*p* < 0.001) while it was not related to age (*p* = 0.794) or gender (*p* = 0.066).

The age, tumor grade, tumor stage, and risk score possessed a strong relationship with OS through univariate Cox regression (Figure 4B). The age, tumor stage, and risk score showed a marked relationship with OS through multivariate Cox regression (Figure 4C). These results demonstrate that the risk score could be an independent prognostic predictor. The age, tumor stage, and risk score were employed to set up a nomogram to display the survival probability rates (Figure 4D). The AUC for 1, 3, and 5 years was 0.864, 0.815, and 0.795, respectively (Figure 4E). The calibration analysis displayed a good fitting effect between the actual survival probabilities and the predicted survival probabilities (Figure 4F).

### 3.4. The FPM Could Be a Convincing Independent Prognostic Predictor in E-MTAB-1980 Cohort

We checked the clinical data and gene expression of the patients in E-MATB-1980 and deleted the cases without complete information, then 92 cases were used to validate the prognostic value of the FPM. According to the media risk score of TCGA, 92 patients were divided into a low-risk group (*n* = 33) and a high-risk group (*n* = 59), and all dead patients were placed into the high-risk group, which illustrated the well-predicted value of FPM (Figure 5A). The two groups of ccRCC patients could be well distributed into two sets by using UMAP and t-SNE analysis (Figure 5B,C). The K–M survival curves of the two risk groups showed that patients in the high-risk group displayed a worse survival rate when no patient was in the low group (Figure 5D). The AUC for 1, 3, and 5 years was 0.888, 0.875, and 0.879, respectively, which showed the convincing prognostic value of the FPM (Figure 5E). The heatmap of the 92 patients in the E-MTAB-1980 cohort was also used to present the relationship between risk group, basic clinical information, pathological features, and nine-gene expression (Figure 5F).

The age, tumor grade, tumor stage, and risk score possessed a strong relationship with OS through univariate Cox regression (Figure 5G). The tumor stage, and risk score showed a marked relationship with OS through multivariate Cox regression (Figure 5H). These results demonstrate that the risk score could be a convincing independent prognostic predictor.

### 3.5. Molecular Functional Analysis

GO and KEGG analysis were used to investigate the molecular function and potential signaling pathways of nine genes. The nine genes enriched in 132 BP, 22 CC, 36 MF, and 10 KEGG terms that possessed significant difference (*p* < 0.05). The top ten terms of BP, CC, and MF were chosen and are displayed in Figure 6A,B. The nine genes mainly focused on several amino acid metabolic processes, metal processes, and redox processes, such as the homocysteine metabolic process, α-amino acid metabolic process, sulfur amino acid metabolic process, ROS metabolic process, electron transport chain, electron transfer activity, oxidoreductase activity, and cellular response to metal iron. The 10 KEGG terms contained HIF-1 signaling pathway, ferroptosis, and amino acid metabolism (Figure 6C,D), however, DPEP1, GLRX5, and CDCA3 did not enrich in 10 KEGG terms.

### 3.6. GLS2 Was Upregulated during Ferroptosis

The erastin or RSL3 could dose-dependently induce the death of ACHN cells that was prevented by the ferroptosis inhibitors Fer-1 and Lip-1 in Figure 7A,B, and the cellular morphology is shown in Figure 7E. In contrast, the autophagy inhibitor HCQ, the necroptosis inhibitor NSA, and the apoptosis inhibitor Z-VAD could not repress erastin- or RSL3-induced cell death. Meanwhile, similar results also occurred in Caki-1 cells (Figure 7C–E). The mRNA expression of nine genes in ACHN and Caik-1 with different treatments are exhibited in Figure 7F,G. The mRNA expression of GLS2 was significantly upregulated in both ACHN and Caki-1 with erastin or RSL3 treatment, which indicated that GLS2 played a crucial role during ferroptosis.

### 3.7. GLS2 Was Low-Expressed in ccRCC Tissues and Closely Related with Prognosis

Compared with normal kidney tissues, the mRNA expression of GLS2 was low-expressed in ccRCC tissues from TCGA (Figure 8A,C), and the proteomic expression of GLS2 was also low-expressed in ccRCC tissues from CPTAC (Figure 8B). In the aspect of histopathological grade, the mRNA expression of GLS2 was lower in G3 and G4 than its counterparts (Figure 8D). In terms of clinical TNM stage from AJCC, the mRNA of GLS2 was low-expressed in Stage III and IV (Figure 8E). The OS, disease specific survival, and progress-free interval illustrated that patients with high GLS2 expression possessed high survival rates compared with their counterparts (Figure 8F–H). These results illustrate that GLS2 was significantly low-expressed in ccRCC tissues and is closely related to the prognosis of ccRCC patients.

### 3.8. GLS2 Might Be a Suppressor of Ferroptosis in ccRCC

As the expression of GLS2 obviously upregulated with treatment of erastin, we knocked down the mRNA expression of GLS2 using shRNA in ACHN and Caik-1 cells (Figure 9A–D and Appendix A). After the knockdown of GLS2, the cell viabilities of ACHN and Caki-1 markedly decreased (Figure 9E,G). The cell viabilities of shRNA groups with erastin treatment were further decreased, and cell death could be repressed by Lip-1. The levels of MDA increased in both ACHN and Caki-1 cells after the knockdown of GLS2 and further reduced in knockdown groups with erastin treatment while the levels of MDA could downregulate by Lip-1 (Figure 9F,H). The BODIPY 665/676 could detect the lipid ROS in cells and the results were consistent with MDA results both in ACHN (Figure 9I) and Caki-1 (Appendix A). As GLS2 participated in the biosynthesis of GSH, the levels of GSH were tested in ACHN and Caki-1 cells (Figure 9J,K). The levels of GSH decreased in shRNA groups and further reduced in shRNA groups with erastin treatment, whereas Lip-1 treatment could not reverse the intracellular GSH level. The intracellular GSH level was detected by MBB staining because MBB could bind with GSH to emit blue fluorescence [31]. The fluorescence intensity of shRNA groups and erastin groups distinctly reduced and did not reverse with Lip-1 treatment in ACHN, which was consistent with GSH detection (Figure 9L). The MBB staining of Caki-1 with different treatments is displayed in Appendix A. These results show that GLS2 might be a suppressor of ferroptosis via affecting biosynthesis of GSH.

## 4. Discussion

Several methods of managing RCC have been utilized such as surgery, ablation, targeted therapy, chemotherapy, and immunotherapy. Although partial or radical nephrectomy and ablation could successfully be applied as ccRCC therapy, 30% of ccRCC patients still develop metastases, which is associated with higher mortality [4]. The 3-year survival rate of patients with nodal invasion is 20–30% after surgery without consideration of T stage [34]. Thermal ablation, cryoablation, and radiofrequency ablation could be taken into consideration for renal masses of less than 3 cm [3]. Targeted therapy, such as VEGF receptor inhibitors and tyrosine kinase inhibitors, has been used for RCC, however, many patients could develop drug resistance after treatment for 6–15 months, especially those with metastatic RCC [35]. These interventions might improve the OS of ccRCC patients, however, complete remission is rare because advanced RCC is a deadly disease.

Ferroptosis has been investigated in different cancers, such as breast cancer [36,37], glioblastoma [38,39], hepatocellular carcinoma [40,41], lung cancer [42,43], and pancreatic cancer [31,44]. A great number of genes, long non-coding RNAs (lncRNAs), and compounds have been studied and explored by researchers and different mechanisms have been reported. Different gene- or lncRNA-based signatures have been explored and possess prognostic value [18,19,45,46,47]. Prognostic models are fundamental to developing a personalized therapy, moreover, an early diagnosis is of paramount importance in these cases [48]. In a recent study, a signature containing eight ferroptotic lncRNAs was found to be accurate and reliable in predicting clinical outcomes, and the target genes (BNIP3, RRM2, and GOT) of three lncRNAs (LINC00460, LINC01550, and EPB41L4A-DT) were closely related to survival outcomes of ccRCC [45]. In the ferroptosis-related gene signature, seven genes were selected to set up a model that showed good prognostic value, and twelve genes were chosen to predict prognosis and reveal immune relevancy, however, the signatures only used a group of 60 ferroptosis-related genes and were only displayed in TCGA with or without simple validation in the E-MTAB-1980 cohort [18,19]. These signatures lack database validation or experimental validation and remain only at the level of computer operation and therefore need deeper investigation. Considering these deficiencies, we introduced three databases from GEO and obtained the whole gene expression, then we used these databases and the KIRC cohort from TCGA to find the common DEGs. In order to avoid the limitation of using only 60 genes, we downloaded the latest ferroptotic gene database (including drivers, suppressors, and markers) without unclassified genes, whose role in ferroptosis is unclear, from FerrDb V2. Based on the five databases, a signature including nine genes was employed to predict outcomes of ccRCC patients. Except MT1G and CA9, the other seven genes of the signature were not involved in previous signatures [16,18,19,46,49,50,51], which might be attributed to using a fragmentary gene set. Therefore, the nine-gene signature in this study could be more successful at distinguishing high-risk and low-risk patients and more accurate in predicting the prognosis of patients.

Except GLS2, the other eight genes involved in the signature could be divided into two groups, the drivers (DPEP1, NOX4, TIMP1, and CDCA3) and the suppressors (MT1G, GLRX5, CA9, and CYBB) of ferroptosis. Dipeptidase 1 (DPEP1) as a membrane-bound glycoprotein could hydrolyze a wide range of dipeptides, and it colocalized with clathrin (endocytic vesicle marker) to induce transferrin endocytosis [52]. Deficiency of DPEP1 could protect kidneys from cisplatin-induced ferroptosis. In kidney samples, DPEP1 expression is strongly related to SLC3A2 that combines with SLC7A11 to form a transport system for cystine. NADPH oxidase 4 (NOX4) could generate intracellular superoxide and promote ferroptosis via oxidative stress-induced lipid oxidation [53]. When NOX4 was inhibited by its inhibitor, cells could display resistance to erastin-induced ferroptosis [8]. Pseudolaric acid B could trigger ferroptosis by activating NOX4 in glioma, and the knockdown of NOX4 made it resistant to Pseudolaric acid B-induced cell death [54]. Metalloproteinase inhibitor 1 (TIMP1) targets and forms complexes with metalloproteinases to irreversibly inactivate the latter. Inhibition of TIMP1 could repress ferroptosis of CMEC cells by decreasing transferrin receptor 1 [55]. The cell division cycle associated protein 3 (CDCA3) mainly participates in drug resistance and cell cycle regulation in cancers and has not been studied in depth [56]. However, CDCA3 was just regarded as a driver of ferroptosis because of genome-wide CRISPR screens and has not been validated [57]. In the other group, metallothionein-1G (MT1G) has a high content of cysteine residues that bind various heavy metals and, as a transcriptional target of NRF2, could ameliorate heavy metals and free radicals to maintain cellular redox homeostasis while it is upregulated in sorafenib-resistant hepatocellular carcinoma cells [41]. Suppression of MT1G expression via shRNA or an inhibitor could significantly improve the sensibility of tumors to sorafenib. Glutaredoxin 5 (GLRX5) participates in iron–sulfur cluster biogenesis and regulates hemoglobin synthesis. GLRX5 knockdown could enhance intracellular lipid peroxidation and increase intracellular free iron, which is attributed to the upregulated transferrin and downregulated ferritin in head and neck cancer cells [58]. Carbonic anhydrase 9 (CA9), as one of the CAs that play a crucial role in equilibrating the reaction between CO_2_, HCO_3_^−^, and H^+^, is inductively expressed during hypoxia in various cancers [59,60]. Inhibition of CA9 could decrease the viability and migration of malignant mesothelioma cells, while Fe^2+^ is increased via upregulating transferrin receptor and downregulating ferritin [61]. The CA9 inhibition could be repressed by deferoxamine and ferrostatin-1, which indicated that CA9 might be a suppressor of ferroptosis. Cytochrome b-245 β chain (CYBB) is the terminal component of a respiratory chain [62]. However, its suppression of ferroptosis was only deduced and not entirely confirmed [8].

There are two types of glutaminase isoenzymes, GLS (encoded by *GLS* and regulated by regulated by c-Myc) and GLS2 (encoded by *GLS2* and regulated by p53), which are both significant enzymes participating in glutamine metabolism. The GLS-mediated deamination of glutamine results in ammonia release to maintain cell survival via biosynthesis with α-ketoglutarate and intermediates, meanwhile, glutamate coming from GLS2-mediated deamination of glutamine takes part in an antioxidant mechanism (GSH) [63]. GLS is correlated with tumor growth rate and malignancy [64], and it is high-expressed in various cancers including brain cancer, lung cancer, breast cancer, hepatocellular carcinoma, colorectal cancer [65,66]. In a recent study, the absence of exogenous glutamine induced glutamate level, which led GLS to convert from dimer to self-assembled filamentous polymer [67]. The catalytic activity of filamentous GLS increased and further depleted intracellular glutamine, which resulted in ROS-induced apoptosis that could be rescued by asparagine supplementation. GLS2 could increase the GSH level to enhance intracellular antioxidant function in HepG2, HCT116, and LN-2024 cells [68]. In our experiments, the viabilities of ACHN and Caki-1 decreased after GLS2 knockdown, and the GSH levels of ACHN and Caik-1 descended accompanied by increase of MDA. According to these findings and our experiments, GLS2 might be a negative regulator of ferroptosis. However, this conclusion was different from another study in which the knockdown of GLS2 repressed serum-dependent necroptosis in mouse embryonic fibroblasts through control of glutaminolysis [69]. The authors also considered that the results might be due to predominant expression of GLS2 in mouse embryonic fibroblasts, however, they did not further validate this. As there are various components in fetal bovine serum, the results lacked specific ferroptosis treatments and specific ferroptosis-related assays. In our study, we used erastin and RSL3 to induce ferroptosis of ACHN and Caki-1 and discovered the role of GLS2 in the ferroptosis of ccRCC cells.

## 5. Conclusions

In summary, this study identified a novel signature that could successfully distinguish patients with ccRCC on the basis of clinical and molecular characteristics. The novel nine-FPEDG signature could be an independent prognostic factor for ccRCC in TCGA and ArrayExpress databases. It was discovered for the first time that GLS2 might be a ferroptotic suppressor in ccRCC. The potential mechanisms of other FPEDGs remain unclear and need further investigation.

## Figures and Tables

**Figure 1 cancers-14-04690-f001:**
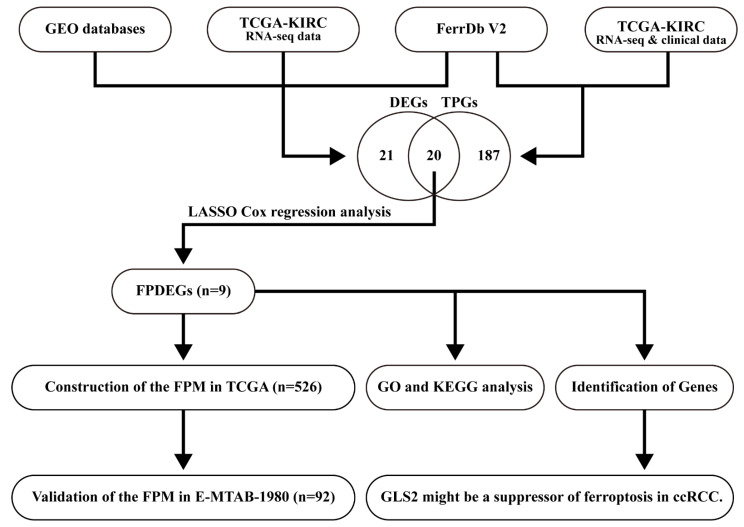
Flow chart of identification and validation.

**Figure 2 cancers-14-04690-f002:**
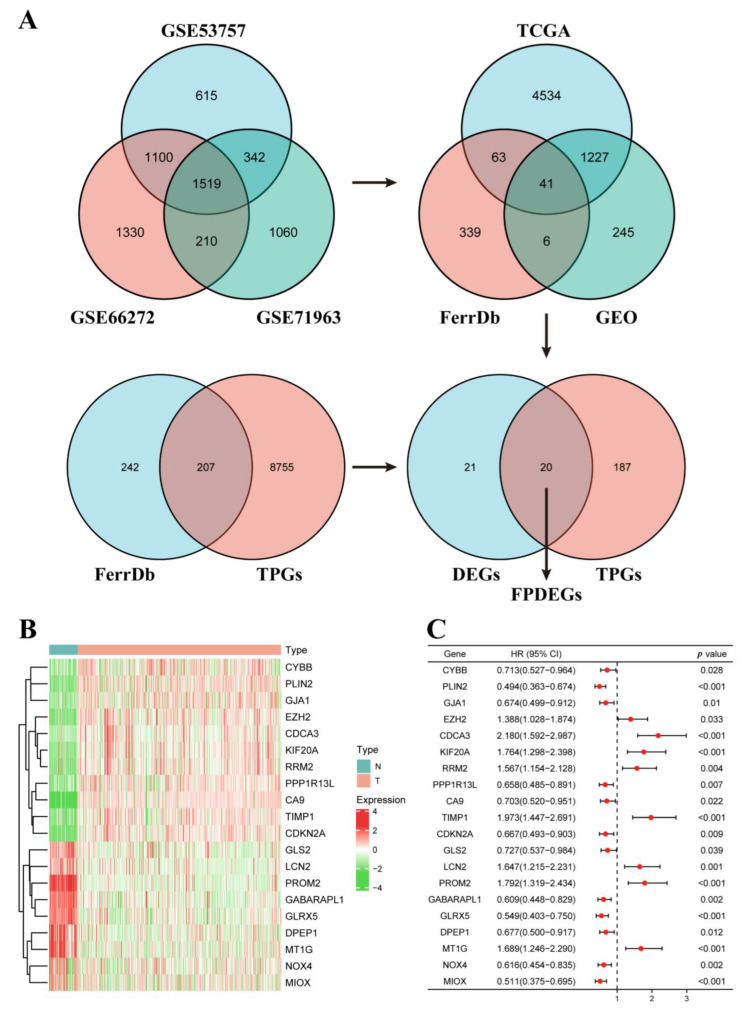
Identification of FPDEGs. (**A**) Process of identifying FPDEGs via Venn. (**B**) Heatmap showing expression of twenty FPDEGs in KIRC cohort of TCGA database. (**C**) Forest plot showing twenty FPDEGs identified by univariate Cox regression analysis based on OS.

**Figure 3 cancers-14-04690-f003:**
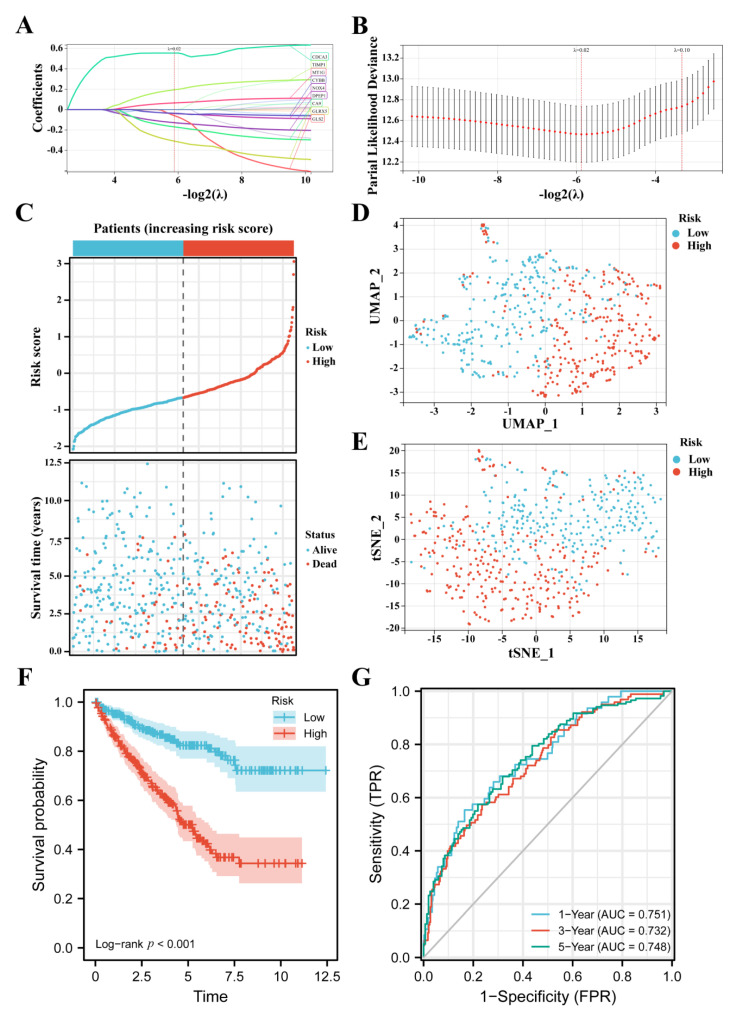
Construction of the FPM in TCGA. (**A**,**B**) LASSO Cox regression analyzed FPDEGs. (**C**) Distributions and media value of ccRCC patients with increasing risk score and distribution of ccRCC patients with corresponding survival status. (**D**) UMAP plot of the ccRCC patients showing the distribution in two risk groups. (**E**) t—SNE plot of the ccRCC patients showing the distribution in two risk groups. (**F**) K–M curves of patients in two risk groups. (**G**) AUC of the time-dependent ROC curves showing the prognostic value of the risk score.

**Figure 4 cancers-14-04690-f004:**
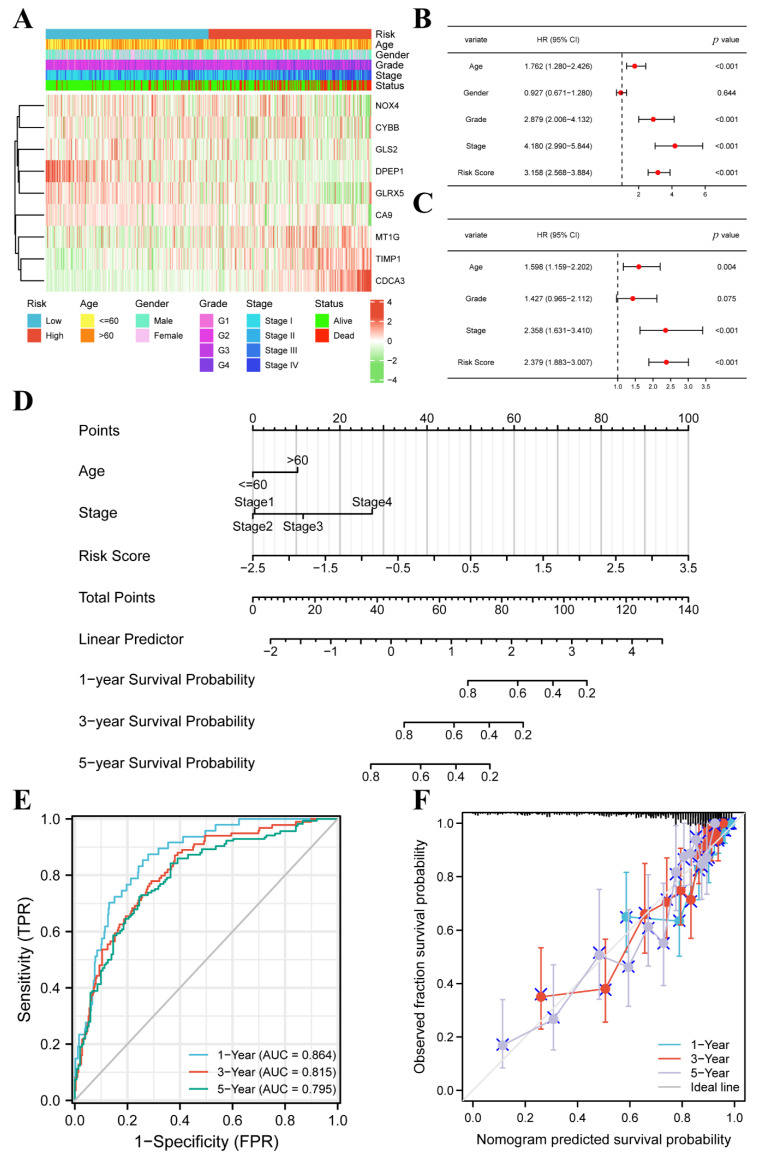
FPM could be an independent prognostic factor for ccRCC. (**A**) Heatmap showing the correlations between risk groups and clinical characteristics. (**B**) Univariate Cox regression analysis of the factors. (**C**) Multivariate Cox regression analysis of the factors. (**D**) A prognostic nomogram model for predicting 1-, 3-, and 5-year survival probability. (**E**) Time-dependent ROC curves of the nomogram. (**F**) Calibration curves of the nomogram for predicting 1-, 3-, and 5-year survival probability.

**Figure 5 cancers-14-04690-f005:**
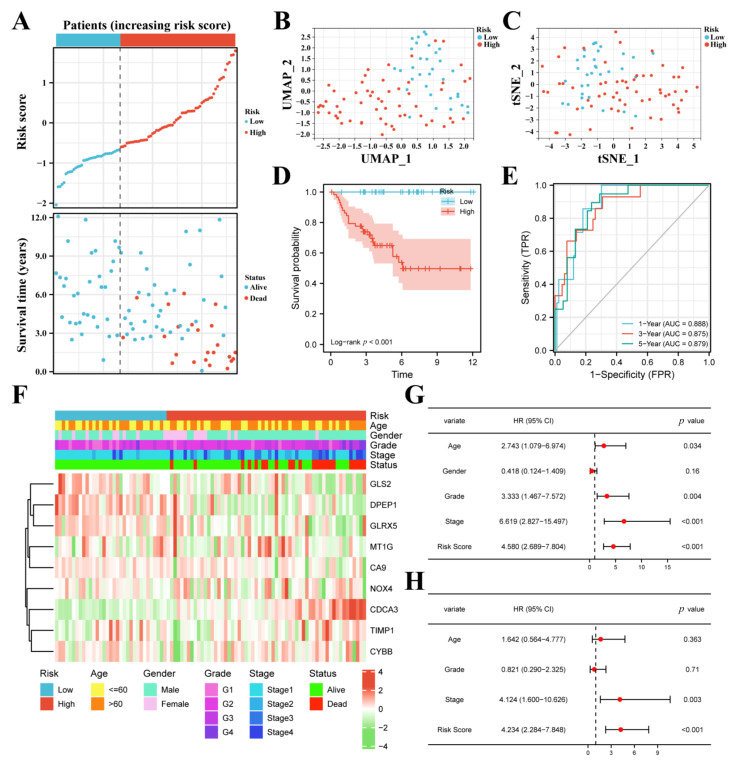
Validation of the FPM in E-MTAB-1980 cohort. (**A**) Distributions and media value of ccRCC patients with increasing risk score and distribution of ccRCC patients with corresponding survival status. (**B**) UMAP plot of the ccRCC patients showing the distribution in two risk groups. (**C**) t-SNE plot of the ccRCC patients showing the distribution in two risk groups. (**D**) K–M curves of patients in two risk groups. (**E**) AUC of the time-dependent ROC curves showing the prognostic value of the risk score. (**F**) Heatmap showing the correlations between risk groups and clinical characteristics. (**G**) Univariate Cox regression analysis of the factors. (**H**) Multivariate Cox regression analysis of the factors.

**Figure 6 cancers-14-04690-f006:**
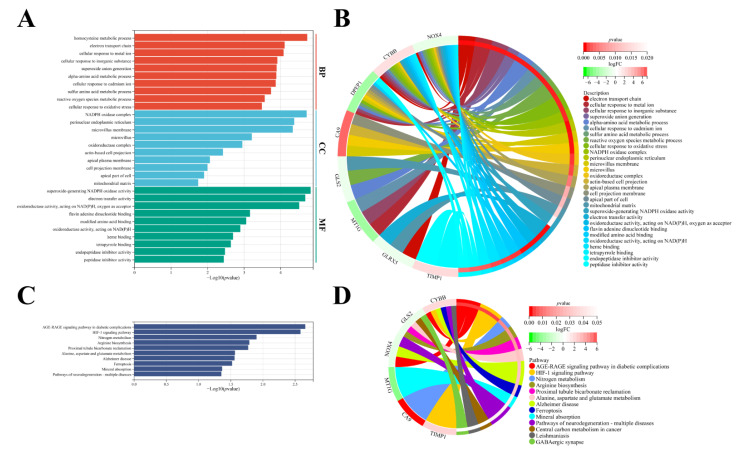
Enrichment analysis of nine FPDEGs. (**A**,**B**) Top ten enriched terms of nine FPDEGs in BP, CC, and MF. (**C**,**D**) Ten KEGG terms of nine FPDEGs.

**Figure 7 cancers-14-04690-f007:**
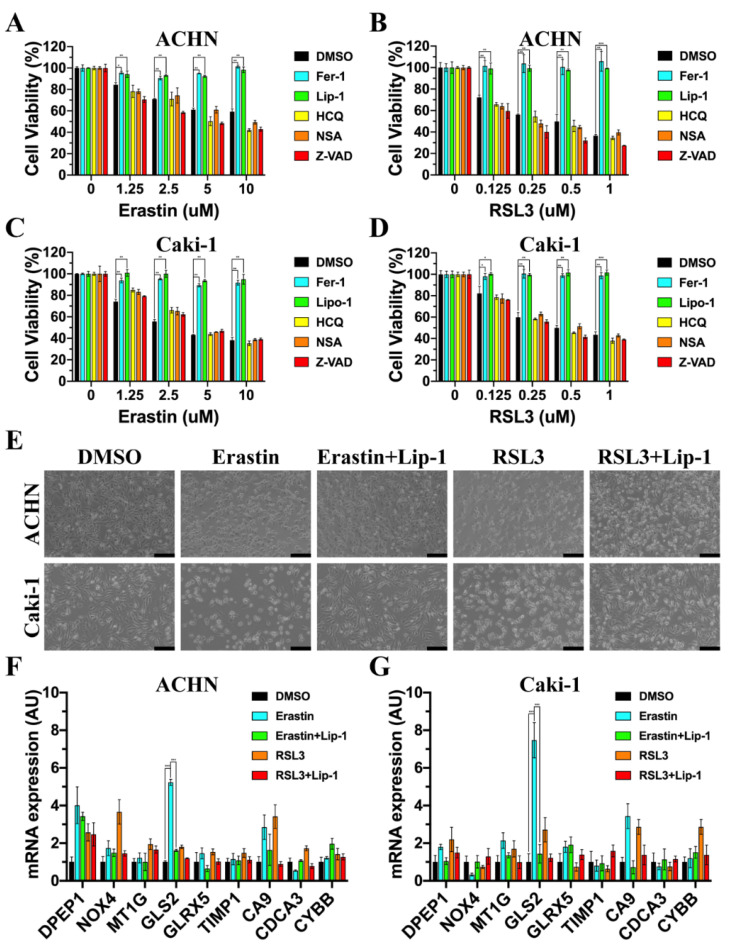
GLS2 was upregulated during ferroptosis. (**A**,**B**) ACHN cells were treated with various concentrations of erastin (12 h) or RSL3 (6 h) with or without Fer-1 (1 μM), Lip-1 (1 μM), HCQ (20 μM), NSA (1 μM), Z-VAD (20 μM). (**C**,**D**) Caki-1 cells were treated with various concentrations of erastin (12 h) or RSL3 (6 h) with or without Fer-1 (1 μM), Lip-1 (1 μM), HCQ (20 μM), NSA (1 μM), Z-VAD (20 μM). (**E**) The cellular morphology of AHCN and Caki-1 treated with erastin (10 μM, 12 h) or RSL3 (1 μM, 6 h) in the absence or presence of Lip-1 (1 μM). The scale bar represented 500 μm. (**F**) The mRNA expression of nine FPDEGs in ACHN treated with erastin (10 μM, 12 h) or RSL3 (1 μM, 6 h) in the absence or presence of Lip-1 (1 μM). (**G**) The mRNA expression of nine FPDEGs in Caki-1 treated with erastin (10 μM, 12 h) or RSL3 (1 μM, 6 h) in the absence or presence of Lip-1 (1 μM). (* *p* < 0.05, ** *p* < 0.01, *** *p* < 0.001).

**Figure 8 cancers-14-04690-f008:**
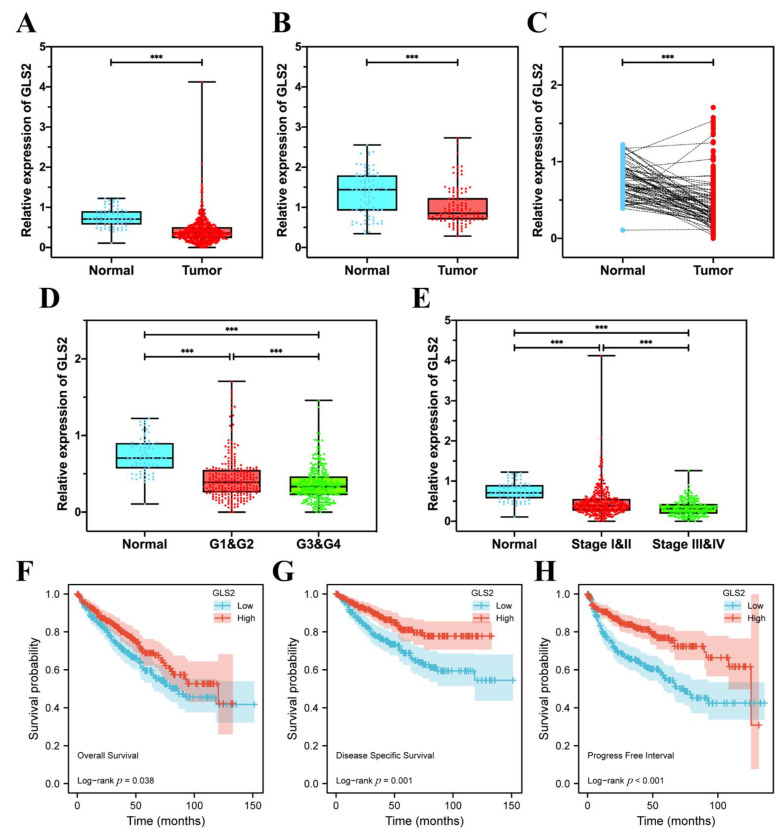
GLS2 was downregulated in ccRCC tissue and closely correlated with prognosis. (**A**) The mRNA expression of GLS2 between normal kidney tissues and ccRCC tissues in TCGA. (**B**) The proteomic expression of GLS2 between normal kidney tissues and ccRCC tissues in CPTAC. (**C**) The mRNA expression of GLS2 between ccRCC tissues and paracancerous tissues in TCGA. (**D**) The mRNA expression of GLS2 in normal kidney tissues and different grade groups in TCGA. (**E**) The mRNA expression of GLS2 in normal kidney tissues and different stage groups in TCGA. (**F**) The OS of ccRCC patients in groups of high or low GLS2 expression. (**G**) The disease specific survival of ccRCC patients in groups of high or low GLS2 expression. (**H**) The progress-free interval of ccRCC patients in groups of high or low GLS2 expression. (*** *p* < 0.001).

**Figure 9 cancers-14-04690-f009:**
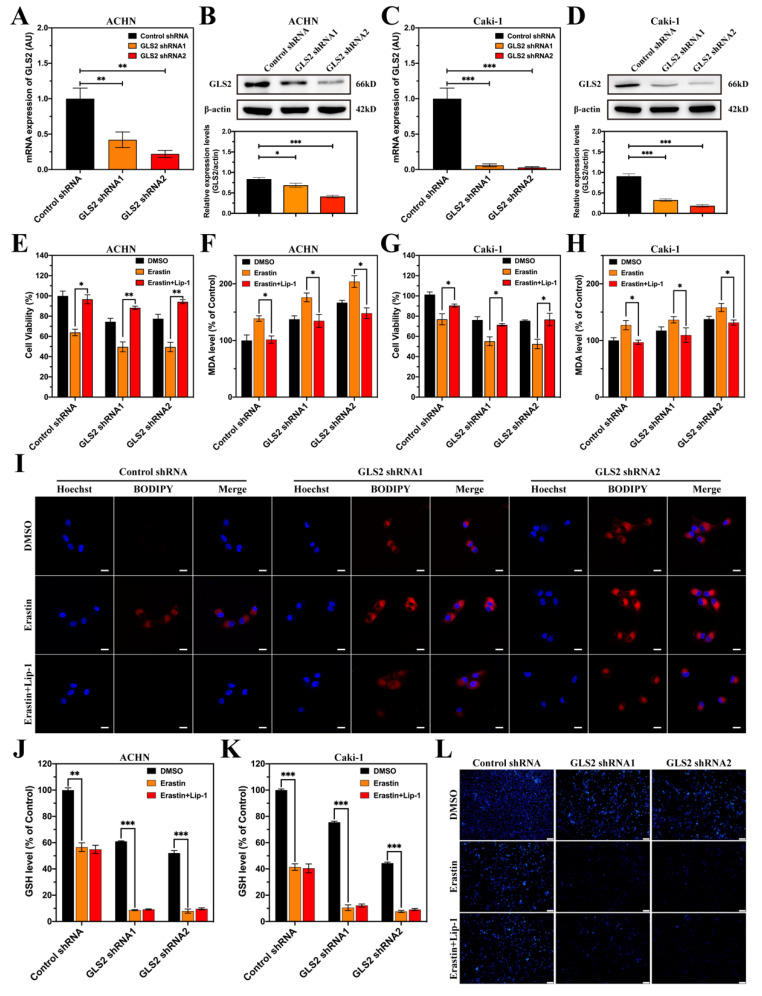
GLS2 might be a suppressor of ferroptosis in ccRCC. (**A**) The mRNA expression of GLS2 with different shRNAs treatments in ACHN. (**B**) The proteomic expression of GLS2 with different shRNAs treatments in ACHN. (**C**) The mRNA expression of GLS2 with different shRNAs treatments in Caki-1. (**D**) The proteomic expression of GLS2 with different shRNAs treatments in Caki-1. (**E**,**F**) The cell viabilities and MDA levels of ACHN treated with erastin (10 μM, 12 h) in the absence or presence of Lip-1 (1 μM). (**G**,**H**) The cell viabilities and MDA levels of Caki-1 treated with erastin (10 μM, 12 h) in the absence or presence of Lip-1 (1 μM). (**I**) Lipid peroxidation of ACHN with various treatments (Nuclei were stained with Hoechst 33342, and lipid peroxides were stained with BODIPY 665/676). The scale bar represented 20 μm. (**J**) GSH levels of ACHN treated with erastin (10 μM, 12 h) in the absence or presence of Lip-1 (1 μM). (**K**) GSH levels of Caki-1 treated with erastin (10 μM, 12 h) in the absence or presence of Lip-1 (1 μM). (**L**) The status of intracellular GSH levels was assessed by MBB staining in ACHN treated with erastin (10 μM, 12 h) in the absence or presence of Lip-1 (1 μM). The scale bar represents 100 μm. (* *p* < 0.05, ** *p* < 0.01, *** *p* < 0.001).

## Data Availability

The data used in this study are available from the corresponding authors.

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
