# Peer review of "Identification and Validation of a Novel Ferroptotic Prognostic Genes-Based Signature of Clear Cell Renal Cell Carcinoma"

_cancers, 2022, doi:10.3390/cancers14194690_

Round 1
Reviewer 1 Report
Please check the attached PDF file

Author Response
Dear reviewer,
Thanks for your suggestions. Here are our revision of manuscript.

Reviewer 2 Report
In this research, the author proposed a ferroptotic prognostic genes-based signature of clear cell renal cell carcinoma (ccRCC). Nine genes were individualized and used in this prognostic model. The crucial gene appeared to be glutaminase 2 (GLS2) which was up-regulated during ferroptosis in ccRCC cells. Cells with GLS2 shRNA displayed lower survival, lower glutathione level, and high lipid peroxide level, which illustrated that GLS2 might be a ferroptotic suppressor in ccRCC.
The paper is well written and a deep analysis has been performed.
I would like to suggest only some revisions:
- In the introduction section please report incidence and mortality only as numbers
- Prognostic models are fundamental to developing a personalized therapy, moreover, an early diagnosis is of paramount importance in these cases (doi: 10.3390/cancers14051112)
Author Response

(The authors gave the same response as above.)
